# Influence of Post-Heat Treatment on Corrosion Behaviour of Additively Manufactured CuSn10 by Laser Powder Bed Fusion

**DOI:** 10.3390/ma17143525

**Published:** 2024-07-16

**Authors:** Robert Kremer, Johannes Etzkorn, Somayeh Khani, Tamara Appel, Johannes Buhl, Heinz Palkowski

**Affiliations:** 1Institute of Metallurgy, Clausthal University of Technology, Robert-Koch-Strasse 42, 38678 Clausthal-Zellerfeld, Germany; robert.kremer@fh-dortmund.de (R.K.);; 2Faculty of Mechanical Engineering, Dortmund University of Applied Sciences and Arts, Sonnenstr. 96, 44139 Dortmund, Germany

**Keywords:** CuSn10, laser powder bed fusion, corrosion, microstructure

## Abstract

This study investigates the influence of heat treatments on the corrosion behaviour of CuSn10 tin bronze, additively manufactured using Laser Powder Bed Fusion (LPBF). LPBF enables the creation of finely structured, anisotropic microstructures, whose corrosion behaviour is not yet well understood. After production, specimens were heat-treated at 320 °C, 650 °C, and in a two-stage treatment at 800 °C and 400 °C, followed by hardness and microstructure analysis. Corrosion tests were conducted using linear polarisation, salt spray, and immersion tests. The results show that heat treatments at 320 °C and 650 °C have no significant effect on the corrosion rate, while the two-stage treatment shows a slight improvement in corrosion resistance. Differences in microstructure and hardness were observed, with higher treatment temperatures leading to grain growth and tin precipitates. The formation of a passive protective layer was detected after 30 h of OCP measurement. Results from other studies on corrosion behaviour were partially reproducible. Differences could be attributed to varying chemical compositions and manufacturing parameters. These findings contribute to the understanding of the effects of heat treatments on the corrosion resistance of additively manufactured tin bronze and provide important insights for future applications in corrosive environments.

## 1. Introduction

Tin bronze alloys, characterised by their excellent combination of corrosion resistance and mechanical strength, are among the oldest and most widely used metal alloys. Traditionally used in areas such as architectural sculptures, offshore components, springs, and bearings, these alloys benefit from their high resilience in most working environments. Typically, tin bronze alloys are processed through casting methods, a long-established technique with a substantial body of knowledge [1,2,3].

With the advent of additive manufacturing, specifically Laser Powder Bed Fusion (LPBF), new possibilities emerged for the fabrication of such alloys. This manufacturing method involves layer-wise deposition of fine powder, locally melted by a laser beam to form individual layers [4]. LPBF, requiring neither tools nor moulds, not only offers significant design freedom but also decouples component complexity and manufacturing effort. The local melting process in the small melt pools results in high cooling rates, leading to a very fine and anisotropic microstructure. This process significantly affects the material, as the strength of additively processed bronze is considerably higher than that of cast material [5,6,7].

CuSn10 is frequently used for additive processing and is known for its good processability [8,9]. After fabrication, the material is not in phase equilibrium but consists of the α-Cu phase (fcc), the metastable β’-Cu13.7Sn phase (bcc), and the intermetallic δ-Cu41Sn11 phase (fcc) [10,11]. This composition results from the high cooling rates combined with the sluggish diffusion of tin. The phase composition can be adjusted through heat treatments: Low-temperature treatments dissolve β’ in favour of δ, whereas high-temperature treatments can lead to a complete transformation to the α-phase [12,13].

Given the economic impact of corrosion and the only partially understood corrosion behaviour of additively manufactured materials, which can vary depending on process-specific heat treatments, further research is essential. It is crucial to determine the impact of heat treatments, as they are often used to adjust mechanical and technological properties. Possible mechanisms through which heat treatments can influence corrosion behaviour include variations in tin content in different phases, which may promote galvanic corrosion, and grain growth due to heat treatments, which could affect the corrosion process [14,15].

## 2. Material and Methods

To investigate the influence of heat treatments on the corrosion behaviour of CuSn10 produced via LPBF, density cubes were first manufactured and heat-treated at various temperatures [12,13]. Their hardness values were then measured, as these correlate with the phase composition. Based on these investigations and various literature references, the actual corrosion specimens were manufactured and heat-treated. A heat treatment of 320 °C for 2 h was expected to result in a microstructure comprising α and δ phases [16]. The heat treatment at 650 °C for 2 h was anticipated to produce a pure α structure [11,12]. The two-stage heat treatment at 800 °C for 2 h, followed by 400 °C for 4 h, was intended to yield a fine-grained structure [16]. The hardness and microstructure were then examined in cross and longitudinal sections. The corrosion behaviour was investigated using accelerated electrochemical methods and non-accelerated immersion and salt spray tests.

### 2.1. Feedstock Material, Manufacturing, Heat Treatment, and Preparation

The starting material for the tests was purchased from m4p material solutions GmbH (Magdeburg, Germany), designation “m4pTM Brz10.07”. According to the works certificate, the material supplied has a tin content of 10.8 wt.% and a permissible proportion of other constituents of 0.5 wt.%. The powder size distribution is shown in Figure 1, while Figure 2a shows an image of the powder. The powder particles predominantly have a spherical shape with few satellites and a uniform size distribution. The powder is therefore well suited for additive manufacturing.

For manufacturing, the LPBF system “Mlab cusing R” by Concept Laser (Lichtenfels, Germany) was used, which employs a 100 W fibre laser with a focus diameter of 50 µm. The utilised manufacturing parameters are based on previous internal investigations and are listed in Table 1.

The scanning strategy is sketched in Figure 2b. The areas to be exposed were divided into 5 × 5 mm^2^ islands and illuminated in a random order, with the laser beam driving a curve at the edges to reduce the number of start and stop points.

The designation of directions and planes is presented in Figure 3 to ensure unambiguous assignment. The cross and longitudinal designations refer to the build direction, which runs along the *Z*-axis.

The specimens were placed into a preheated furnace for heat treatment and cooled outside of the furnace in ambient air at the end of the treatment time. To generally investigate the effect of heat treatment temperature, specimens were treated from 200 °C to 1000 °C in 100 °C steps. The subsequent heat treatments for the corrosion studies are derived from various literature sources. The treatments were performed at 320 °C for 2 h, 650 °C for 2 h, and 800 °C for 2 h followed by 400 °C for 4 h.

Cubes with an edge length of 10 mm were produced to analyse the hardness and microstructure. The hardness was analysed using the Vickers small-force hardness test. Measurements were taken using an equilateral diamond pyramid with an opening angle of 136°, applying the test force of 10 kp (1 kp = 9.80665 N) for 15 s. The measuring and counter surfaces were pre-polished with P1000 grit SiC (Short for “silicon carbide”) wet sandpaper prior to the measurements. The hardness was always measured both longitudinally and cross-sectionally, with five measurements being taken in each direction. These five measurements were evenly distributed over two opposite faces of the cube to ensure the minimum permissible distance during measurement. A total of 10 individual measurements were therefore carried out for each specimen.

To prepare the specimens for the microstructure investigation, the cubes were hot embedded in bakelite with a top layer of epoxy resin. The specimens were prepared through several grinding and polishing steps (grinding: 80, 180, 320, 600, 1200, and 2500; polishing: 3 µm, 1 µm, and OPS) and then, as required, treated with aqueous etching solutions of ammonia and hydrogen peroxide or iron (III) chloride and hydrochloric acid. The as-built specimen was etched for 35 s using ammonia and hydrogen peroxide etching solutions, whereas the 320 °C for 2 h specimen required 120 s. The etchant had no effect on the specimens treated at higher temperatures. For these, a solution of iron (III) chloride and hydrochloric acid was used; the specimen treated at 650 °C for 2 h was etched for 4 s and the one treated at 800 °C for 2 h followed by 400 °C for 4 h was etched for 6 s. For the other two specimens, this solution resulted in immediate over-etching, even after multiple dilutions.

Specimens of size 25 × 25 × 2 mm^3^ were used for the corrosion tests.

### 2.2. Examination Methods

Due to the complex nature of corrosion studies, multiple testing methods were implemented to ensure a comprehensive analysis of the corrosion characteristics. Specifically, Linear Sweep Voltammetry (LSV) was employed to characterise the electrochemical behaviour of the CuSn10. This method allows the determination of the corrosion potential and rate, as well as the examination of the alloy’s passivation tendency. However, it is important to note that the method employs highly accelerated conditions, which may lead to deviations from the actual corrosion behaviour. For more in-depth investigations, both salt spray tests and immersion tests were conducted. These tests determine the corrosion rate through the mass loss of the specimens. The salt spray test simulates accelerated corrosion conditions typically found in salty or maritime environments and is often used for industrial quality control [17]. The ASTM Committee G-1 on Corrosion of Metals, responsible for the salt spray standard ASTM B117, notes that the results of the salt spray test rarely correlate with the behaviour in natural environments [18]. The immersion test evaluates the long-term durability of CuSn10 in specific corrosive media. By continuously exposing the material to a corrosive solution over an extended period, realistic data can be gathered [19]. By combining these three methods, corrosion effects across individual specimens can be reliably detected. As the aim of this study is to detect changes due to heat treatments, the corrosion tests do not need to be adapted to any specific application. Accordingly, the methods are performed under standard parameters to enable as high comparability as possible with other studies.

In LSV, the change in the electrical potential of an electrode (in this case the material specimen) is increased linearly with time compared to a reference electrode, while the resulting current is measured. This method makes it possible to obtain information about the corrosion behaviour of the specimen. For the investigation, the specimens were processed with P2500 grit SiC wet sandpaper after the respective heat treatment. The corresponding measurements were carried out with the galvanostat/potentiostat Autolab PGSTAT204 from Metrohm (Filderstadt, Germany). A three-electrode arrangement was used with a secondary calomel electrode as the reference electrode and a platinum electrode as the counter electrode. A circular specimen surface of 100 mm^2^ was demarcated with a PTC1 coating tape from Gamry International (Warminster, PA, USA) and exposed to the electrolyte (NaCl, 3.5 wt.%) at 30 °C in a thermostatically controlled cell. Four measuring surfaces were prepared for each specimen permutation. On the first, the open circuit potential (OCP) was recorded for 60 min in order to estimate the time until a stable state was reached. Based on this preliminary investigation, the OCP was measured at each of the remaining three measuring points to ensure stable conditions at the interface between the specimen and the electrolyte and then the actual LSV measurement was carried out. A potential sweep of −0.5 to 1.5 VSCE (volt versus saturated calomel electrode) with a scan rate of 0.5 mV/s was measured in relation to the respective OCP. Based on the findings in [20], the investigations were carried out with an OCP recording time of 60 h in order to observe the formation of the passive layer due to the change in the OCP, on the one hand, and to investigate the behaviour of the passivated surface at LSV, on the other.

The OCP and LSV measurements were recorded and analysed using the Nova software version 2.1.5 from Metrohm in Filderstadt, Germany. The characteristic values *E*_corr_ (corrosion potential) and *I*_corr_ (corrosion current) were determined from the current versus potential graphs obtained from LSV measurements through Tafel analysis, utilising manual slope determination with the “Perform fit” feature for support. At a specified density of *ρ* = 8.74 g/cm^3^ [21] and a molar mass of *M* = 69.062 g/mol [22], the software calculates the corrosion rate *CR* with Equation (1) [23]. The required corrosion current *I*_corr_ was calculated using the Butler–Volmer equation [23].
(1)$$CR=\frac{3.17}{10^{9}}\;\frac{M}{nF\rho A}\;I_{corr}$$

*n*: Number of electrons exchanged in the reaction.*F*: Faraday constant.*A*: Exposed surface area.

The salt spray test was conducted according to ASTM B117–11 [17] at 35 °C using a 5% NaCl solution for 336 h. The specimens were previously prepared with P120 grit wet sandpaper. As an unaccelerated method for investigating corrosion properties, an immersion test was carried out according to ASTM G31−21 [19]. The surface of the specimens was prepared with P120 grit wet sandpaper and examined in a 3.5 wt.% NaCl solution at 35 °C. The test setup is sketched in Figure 4. Each specimen was tested standing upright in a separate glass beaker containing 500 mL of NaCl solution, covered with a watch glass. All the beakers were positioned together in a thermostatically controlled tank with a random arrangement water tank for temperature control.

The duration of the immersion test was set to 21 days on the basis of the estimation Equation (2) [19] and the results of an LSV study from [24].
(2)$$Duration\;of\;test\;\left(h\right)=\frac{50}{corrosion\;rate\left(\frac{mm}{y}\right)}$$

Before and after the corrosion tests using the salt spray and immersion test, the specimens were weighed using a Sartorius laboratory balance with a graduation value of 0.1 µg. Prior to weighing after the corrosion test, the specimens were rinsed with fully mineralised water and any loosely adhering deposits were removed with a soft brush. The mass loss was converted from mass loss per area per unit of time to length per unit of time using a density of 8.74 g/cm^3^ [21]. This was then annualised to express the rate over the course of a year.

To assess the significance of differences in corrosion resistance due to heat treatment, two-sided independent specimens t-tests are conducted with a significance level of α = 0.05. Significant results are denoted with an asterisk (*). These tests compare the as-built condition with the heat-treated states in terms of the metrics open circuit potential (OCP) and corrosion rate (*CR*).

## 3. Results

### 3.1. Microstructure and Hardness

Figure 5 shows the microstructure in both the transverse and longitudinal sections before and after the individual heat treatments. As the treatment required for etching the different specimens was very different, it can be deduced that the heat treatments significantly influenced the chemical reaction behaviour of the material. In the as-built condition as well as after low-temperature heat treatment, the typical structure of LPBF manufactured materials is recognised.

In the longitudinal section, the individual fusion tracks and their penetration depth can be clearly recognised, while the influence of the intersecting tracks can be seen in the cross section. The grain structure is hardly recognisable due to its fineness. Thus, the as-built material is also shown in an SEM image with higher magnification in Figure 6a, wherein the fine grains at the transitions of the individual melt tracks are visible. In Figure 5, the difference between the cross and the longitudinal plane is clearly noticeable for the as-built material, indicating the anisotropic state of the material, which was not visibly affected by the heat treatment at 320 °C for 2 h. The overall appearance changed significantly after the two high-temperature treatments: The cross and the longitudinal section cannot be visibly distinguished anymore. Furthermore, the typical appearance of LPBF manufactured materials is no longer recognisable. Accordingly, a decrease in anisotropy is expected. After treatment at 650 °C for 2 h, a homogeneous microstructure is recognised, while after the two-stage heat treatment at 800 °C for 2 h + 400 °C for 4 h, precipitation occurs, dominated by tin. An Energy Dispersive X-ray spectroscopy (EDX) measurement yielded a tin content of approximately 20 wt.%. To better recognise the precipitation, a magnified SEM image of one of the precipitates is depicted in Figure 6b. Notably, large pores appear after the heat treatment, which are located at the grain boundaries. Generally, an increasing grain growth with the temperature of the heat treatment is observed.

The determined hardness values show that the component orientation has no influence on the hardness. Consequently, a combined value (mean) will be used henceforth. The detailed summarised values are shown in Appendix A. The specimen treated at 900 °C was partially melted, and the specimen treated at 1000 °C was almost entirely melted. The hardness values are visualised in Figure 7, with the comparison values displayed in grey and the corrosion specimens colour coded. This allows for a good assessment of the specimens’ phase composition. For the additively manufactured specimens, a significant change due to heat treatments is noticeable. The hardness increases up to 320 °C, which could be due to a transformation of β’-Cu13.7Sn into δ-Cu41Sn11—a behaviour that was already described in [12,13]. The subsequent drop in hardness values up to α-Cu is due to the transformation from β’-Cu13.7Sn to α-Cu. Beyond 700 °C, no clear trend can be described [12,13]. The fluctuations observed here can be attributed to precipitations and the onset of melting.

Since all specimens were made from the same powder, the influence of the manufacturing method is clearly discernible.

### 3.2. Electrochemical Corrosion Testing

The course of the potential during the open-circuit potential measurement can be seen in Figure 8. For the CuSn10 specimens, a steady state can be assumed from 900 s onwards. Accordingly, the time for the OCP measurement is set to 900 s in the following LSV measurements.

Figure 9 shows the CuSn10 specimens in the four examined states, comparing the X/Y section (blue) with the respective Z section (yellow), and in no case could a difference be detected. Consequently, no anisotropy in corrosion behaviour could be demonstrated. The average OCP of all specimens after 900 s is −0.245 (±0.001) V.

For comparing the conditions and materials between each other, the graphs were averaged into a single graph per specimen in Figure 10, regardless of the orientation. Since no dependency on orientation could be determined, this approach is not expected to result in any loss of information. The additively manufactured CuSn10 specimens show no differences between each other.

The complete values of the LSV measurement are listed in Table 2, including the significance analysis results. Neither a change in the open circuit potential nor in the corrosion rate can be stated.

### 3.3. Electrochemical Corrosion Testing—Long-Time OCP

The progression of the OCP measurements over 60 h is plotted in Figure 11. In general, the specimens exhibit similar behaviour, indicating that the heat treatments have no significant influence. The graphene increases by approximately 70 mV in the first 30 h and then stabilises, which can be explained by the formation of a passive layer. Fluctuations in the graphs can be attributed to local effects on the measuring surface, such as the detachment of corrosion products. The behaviour of the specimens heat-treated at 800 °C for 2 h followed by 400 °C for 4 h is noteworthy as they show a more pronounced initial drop compared to the other specimens, but level off after approximately 4 h, following the overall trend. The average OCP after 60 h for all specimens is −0.180 (±0.009) V.

The graphs of the LSV measurement after 60 h of OCP are depicted in Figure 12. On the left side, the complete progression is shown. On the right side, the transition area between cathodic and anodic is enlarged. The fluctuations in the graphs are noticeable, and no clear corrosion potential or corrosion current can be identified. A potential explanation is the formation of inhomogeneous passive layers, leading to local differences in reactivity. The detachment of individual parts of the passive layer can result in jumps in the graph. Due to the course of the graphs, Tafel analysis to determine the corrosion rate is not feasible. However, it is noteworthy that the specimens heat-treated at 800 °C for 2 h followed by 400 °C for 4 h exhibit a significantly more pronounced transition than the other specimens, indicating a more uniform passive layer.

### 3.4. Electrochemical Corrosion Testing—Chemical Composition of the Test Surface

The chemical composition of the corroded surface was measured using EDX and compared to the base material. The measurement results are listed in Figure 13 and Table 3. Notably, the tin content measured on all corrosion surfaces, compared to the base material, indicates the formation of tin oxides. Additionally, a significant increase in chlorine was recorded at 800 °C for 2 h + 400 °C for 4 h, suggesting additional formation of chlorides.

The emerging passive layers consist of copper and tin oxides. Only in the specimens treated at 800 °C for 2 h + 400 °C for 4 h do chlorides form in detectable amounts.

### 3.5. Gravimetric Corrosion Testing—Immersion and Salt Spray Test

The results of the gravimetric corrosion tests are listed in Table 4, including the p-value of the t-test. Significant changes are marked with an *. Both tests demonstrate corrosion rates of the same order of magnitude. In both tests, the specimens treated at 320 °C for 2 h and 650 °C for 2 h show no significant difference compared to the as-built specimen. The specimens treated at 800 °C for 2 h + 400 °C for 4 h exhibit an average reduction in corrosion rate of 21% in the salt spray test and 15% in the immersion test. However, a statistically significant difference is observed only in the immersion test. It should be noted that the values for the specimens treated at 800 °C for 2 h + 400 °C for 4 h in both the immersion test and salt spray test are close to the significance threshold. Therefore, within the current test setting, it cannot definitively be determined whether there is a minor difference in the corrosion rate or no difference at all.

## 4. Discussion

A significant influence of the heat treatments on the CuSn10 was found. Treatment at 320 °C for 2 h did not change the visual impression of the metallographically produced microstructure, but did lead to an increase in hardness. The clear difference between with and against the build-up direction in the micrograph is still clearly visible, which is why a reduction in anisotropy cannot be assumed.

This change can be explained by a phase transformation in which the β’ phase in the microstructure of α, β’, and δ transforms into δ. The δ phase is significantly more brittle [12,13]. In addition, the specimen required a significantly longer exposure time to the etchant compared to the untreated specimen.

The heat treatment at 650 °C for 2 h caused a significant change in the microstructure. From the originally fine, process-typical microstructure with visible melting traces, uniform grains formed, indicating significant grain growth. A preferred direction was no longer discernible, suggesting the reduction in anisotropy. The hardness of the specimen decreased, indicating a transformation in the δ-phase into the α-phase, a process that was documented several times in the literature [11,12,20].

In the treatment at 800 °C for 2 h followed by 400 °C for 4 h, precipitations with increased tin content formed at the grain boundaries, leading to a slightly increased hardness level. Such precipitations are known from casting and often consist of α- and δ-phases [25]. Additionally, no anisotropy was recognisable and further grain growth occurred.

The conducted investigations to assess corrosion behaviour are only partially comparable, as different salt concentrations and exposure conditions as well as accelerated and unaccelerated methods were used. Nevertheless, all results—as shown in Figure 14—fall within the same corrosion protection category (Level 2; material loss 0.1 to 0.5 mm/year) for the classification of copper materials according to the German Copper Institute (DKI). The DKI classifications for copper materials were chosen because they are more detailed than the more well-known DECHEMA table for corrosion progress. According to DECHEMA, the tested material would fall into class + (material loss 0.1 to 1 mm/year) [26,27]. This indicates that the heat treatments and the differences in the tested environmental conditions only result in differences that are technically insignificant.

Information on the passivation process is provided by comparing the LSV measurement after 900 s with the LSV measurement after 60 h. The OCP after 60 h is −0.180 V instead of −0.245 V as observed after the 900 s measurement. This increase can be explained by the formation of a passivation layer and its protective effect. Although it was not possible to determine the corrosion parameters on the passivated layer, the comparison with the LSV measurements at 900 s OCP in Figure 15 shows their influence. The corrosion currents are generally lower, indicating a reduced corrosion rate, and the repeatability is significantly reduced, suggesting an uneven formation of the passivation layer.

The measurements indicated that heat treatments at 320 °C for 2 h and 650 °C for 2 h have no impact on the corrosion rate. All tests showed no difference compared to the as-built condition. Accordingly, the detected changes in microstructure have no effect on the corrosion resistance in a saline environment.

Specimens treated at 800 °C for 2 h + 400 °C for 4 h displayed an anomaly during the 60 h OCP measurement. This suggests that the passivation process might differ. This is also supported by the chemical composition of the measurement surface, where only the specimens treated at 800 °C for 2 h + 400 °C for 4 h showed a higher chlorine content. Furthermore, these specimens exhibited a trend towards higher corrosion resistance in the gravimetric corrosion tests, although this was only significant in the immersion test. It should be noted that the specimen size used is only somewhat suitable for detecting such fine differences. Since differences between the specimens treated at 800 °C for 2 h + 400 °C for 4 h and the as-built condition appeared in several different tests, a slight difference can be expected, which is not technically relevant under the tested environmental conditions.

In combination with the microstructure analysis and the direction-dependent LSV measurement, it suggests that changes in phase composition, grain size, and orientation do not affect the corrosion behaviour under the tested conditions. Only the tin-containing precipitates seemed to influence the corrosion behaviour, although this influence is not technically significant. This behaviour cannot be easily generalised to other media, as indicated by observations during etching, where resistance to etching agents generally increased due to the heat treatments.

Classification within the field of research:

The influence of heat treatments on hardness as shown in [12,13] could be reproduced in this study. Notably, in [16], after heat treatments at 800 °C for 2 h and 400 °C for 4 h, the microstructure of the specimen changed from a columnar grain to an equiaxed grain with numerous annealing twins. The same heat treatment in the present study led to significant precipitations at the grain boundaries. The reported tin contents of 11.01 wt.% in [16] compared to 10.8 wt.% in the present study cannot account for the difference. One possibility for the discrepancies could be the allowable 0.5 wt.% of other elements from the supplier in the present study. Along with potential unreported additional elements in [16], this could result in a sufficient difference in the chemical composition of the materials.

Comparisons between different corrosion behaviour studies are often difficult. This is because corrosion tests are often set up differently to reproduce various conditions, as many factors can influence corrosion behaviour. In addition, there is the influence of the different manufacturing parameters in the LPBF process, as well as variations between material batches and suppliers.

In [24], the corrosion behaviour was investigated using LSV measurements on five specimens. A standard three-electrode system was employed, utilising a GAMRY Interface 1000 (UK) potentiated in a 3.5 wt.% NaCl solution. A Saturated Calomel Electrode (SCE) served as the reference electrode, while a graphite electrode was used as the counter electrode. Similarly, the corrosion behaviour in the as-built condition was investigated in [28] using a comparable setup, within a measurement potential range from −0.25 to 1.5 VSCE and a scan rate of 0.5 mV/s, also in a 3.5% NaCl solution. In [20], the corrosion behaviour of LPBF-manufactured CuSn10 was examined both in the as-built condition and after heat treatments at 600 °C for 1 h and 800 °C for 1 h. In each condition, three specimens with polished surfaces were tested. During the LSV investigations, the open circuit potential was recorded for 60 h, followed by an LSV measurement in the potential range from -1.5 to 0 VSCE with a scan rate of 1.67 mV/s. A standard three electrode corrosion cell was used for the tests, featuring a saturated calomel electrode (SCE) as the reference electrode, a platinum counter electrode, and a CHI-604C corrosion tester (CH Instruments, Inc., Bee Cave, TX, USA).

For a comprehensive between-study assessment, *I*_corr_ and *E*_corr_ are plotted and compared in Figure 16, as not all studies determined a corrosion rate.

*E*_corr_ and *I*_corr_ from [24,28] are comparable with the values of the present study, which is also reflected in the comparable corrosion rate of 0.111 mm/year (indicated in the paper: 4.388 mpy). The results from [20] show substantially differing values. In the as-built condition, a considerably higher *I*_corr_ and a markedly lower *E*_corr_ were measured compared to both [24] and the present study. Furthermore, [20] demonstrated a significant change in corrosion behaviour due to thermal post treatment, which could not be replicated here, neither with 900 s of OCP measurements nor, as in [20], with 60 h of OCP measurements. The experimental setups are comparable; however, there are notable differences in the scan rate (1.67 mV/s in [20] versus 0.5 mV/s in this study) and the measurement range (−1.5 to 0 VSCE in [20] versus −0.5 to 1.5 VSCE in this study). When comparing the 60 h OCP measurements between [20] and the present study, it is apparent that significant differences already emerge, which can only be attributed to the test material itself. The as-built material investigated in [20] behaves similarly to all the specimens measured here, whereas the heat-treated specimens in [20] start the measurement with a slightly higher OCP and reach a stable potential after approximately 8 h (instead of 30 h). This suggests a difference in the test material, such as varying chemical composition or influences from the LPBF manufacturing process. Although this does not exclude the impact of the LSV measurement settings, it indicates that the material itself differs.

In [20], alongside electrochemical measurements, a gravimetric method was also used to investigate corrosion. For this purpose, three specimens were weighed and immersed in a 3.5 wt.% NaCl solution for 24 h. After drying and cleaning, they were weighed again. This process was repeated a total of eleven times. The individual data points were approximated using a linear function and used to determine the corrosion rate in terms of mass per unit area per unit time. Using a density of 8.74 g/cm^3^ [21], the following corrosion rates can be calculated: 0.256 mm/year (*R*^2^ = 0.98) (indicated in the paper: 0.612 mg/cm^2^·day) for the as-built state, and for the treatments at 600 °C and 800 °C, corrosion rates of 0.196 mm/year (*R*^2^ = 0.96) (indicated in the paper: 0.469 mg/cm^2^·day) and 0.195 mm/year (*R*^2^ = 0.96) (indicated in the paper: 0.469 mg/cm^2^·day), respectively. Similarly to the electrical chemical tests in [20], the corrosion resistance improved with heat treatments, which could not be reproduced in the present study. Given the limited influence of the experimental setup, the difference between [20] and the present study can be attributed to the tested material. Generally, the corrosion rates calculated in [20] fall within the same corrosion class (corrosion rate: 0.1 to 0.5 mm/year) [27] as all other measurements investigated.

## 5. Conclusions

Corrosion investigation in saline solution

All corrosion rates measured in this study as well as those determined in the literature are in the range of 0.1 to 0.5 mm/year, regardless of measurement method and post treatment.The heat treatments of 320 °C for 2 h and 650 °C for 2 h showed no effect on the corrosion rate in both accelerated and non-accelerated corrosion measurements. The heat treatment of 800 °C for 2 h followed by 400 °C for 4 h showed a tendency to improve the corrosion rate in both immersion and salt spray tests, although the specimen size (n = 5) is not sufficient to make a definitive statement.The results could reproduce the measurements in [24], while those measured in [20] could not be reproduced. The reasons for the differences cannot be stated with certainty but appear to occur independently of the measurement procedure.Parallel to the investigations in [20], no correlation could be detected between build direction and corrosion behaviour.The material forms a protective passive layer, which exhibits a lower open-circuit potential. The formation of this layer took approximately 30 h.

Microstructure and hardness:

The hardness increased from approximately 160 HV 10 with increasing heat treatment, reaching a maximum of about 190 HV 10 after 1 h at 300 °C, and then decreased to about 100 HV 10 at higher heat treatment temperatures.The macrostructure visible in the polished section showed no change after treatment at 320 °C for 2 h. Treatment at 650 °C for 2 h resulted in clearly visible grain growth, and the visible differences in the cross and longitudinal section were no longer discernible. Treatment at 800 °C for 2 h followed by 400 °C for 4 h led to visible tin-containing precipitates.The microstructure resulting from the heat treatment of 800 °C for 2 h followed by 400 °C for 4 h observed in [16] could not be reproduced in this study, which may be attributed to differences in chemical composition. Such differences may arise both in the starting material and during the manufacturing process itself. During the LPBF process, elements can evaporate from the melt, potentially altering the chemical composition. This variable is influenced by the combination of machine, material, and operator, due to different process parameters such as laser power or scanning speed, leading to variations in temperature profiles and melt pool dimensions.

In the extensive corrosion investigations conducted in this study, no impact of heat treatments on corrosion behaviour was detected. However, since such influences were demonstrated in other studies, future research should aim to identify the causes of these differences in the material. It is important to determine whether these differences are due to the chemical composition or to the effects of additive manufacturing, which may occur only with certain combinations of manufacturing parameters.

## Figures and Tables

**Figure 1 materials-17-03525-f001:**
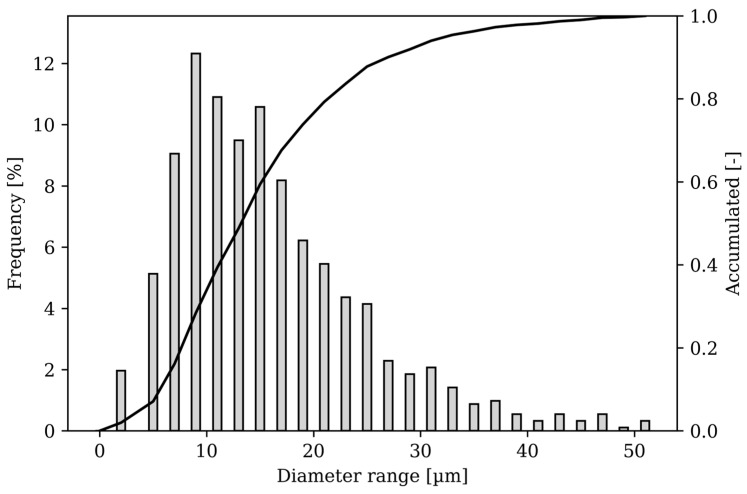
Particle size distribution of the CuSn10 powder used for LPBF.

**Figure 2 materials-17-03525-f002:**
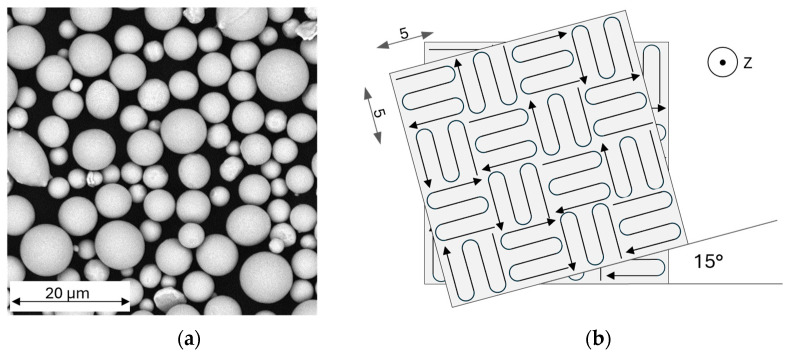
(**a**) The SEM image showing the morphology of the source powder material and (**b**) the scanning strategy, including the size of the island and rotation angle.

**Figure 3 materials-17-03525-f003:**
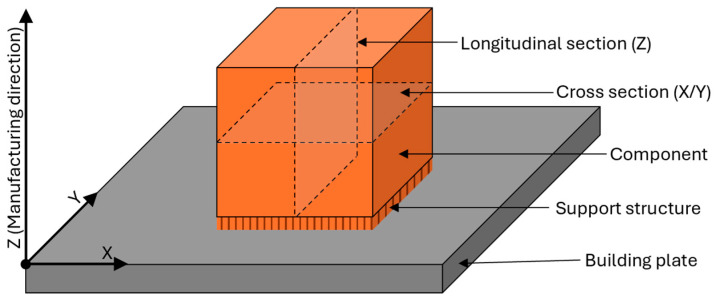
Nomenclature of the axes and planes in the investigations.

**Figure 4 materials-17-03525-f004:**
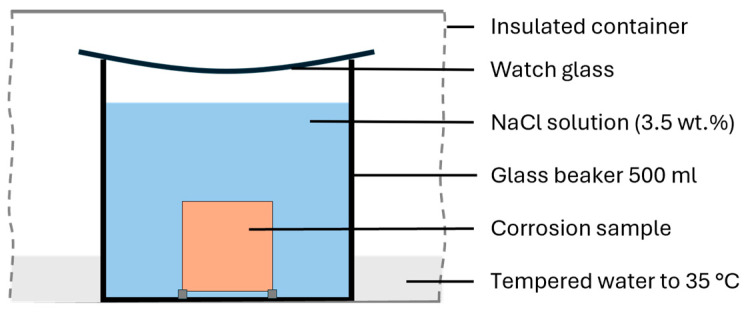
Sketched structure of the immersion corrosion testing.

**Figure 5 materials-17-03525-f005:**
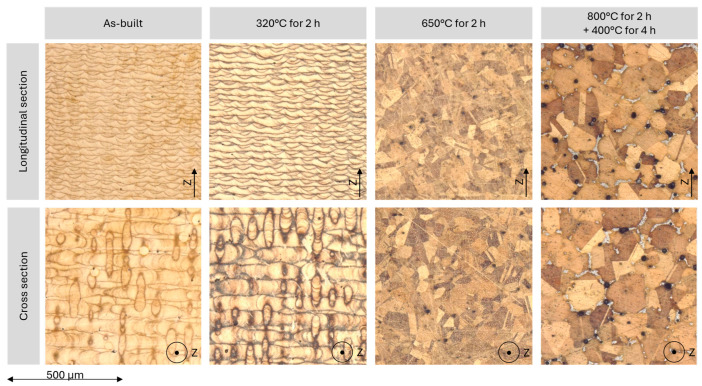
Metallographically prepared micrographs of the specimens in cross and the longitudinal section.

**Figure 6 materials-17-03525-f006:**
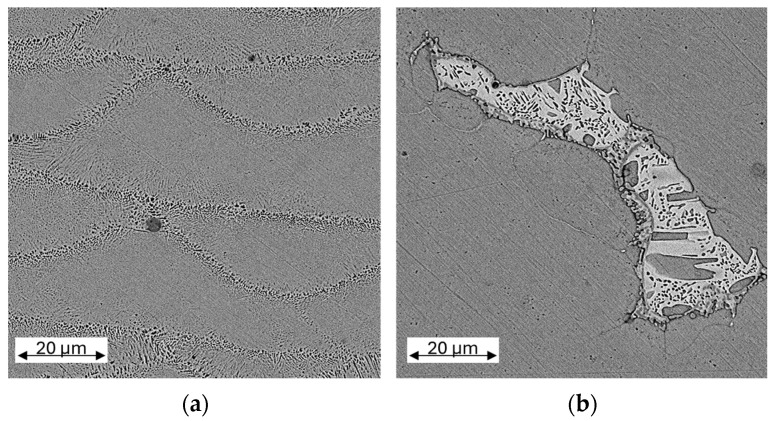
SEM images of the microstructure. Left (**a**) shows the as-built specimen and right (**b**) shows one of the precipitates of the specimen after 800 °C for 2 h + 400 °C for 4 h heat treatment.

**Figure 7 materials-17-03525-f007:**
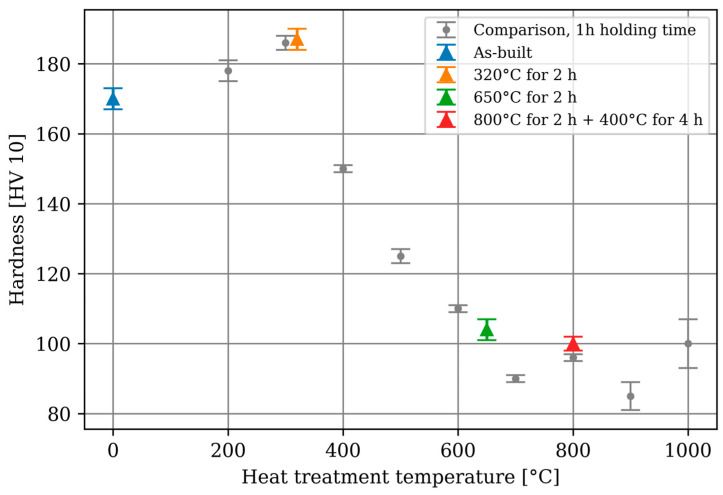
Hardness values of the analysed specimens. Mean and standard deviation are depicted.

**Figure 8 materials-17-03525-f008:**
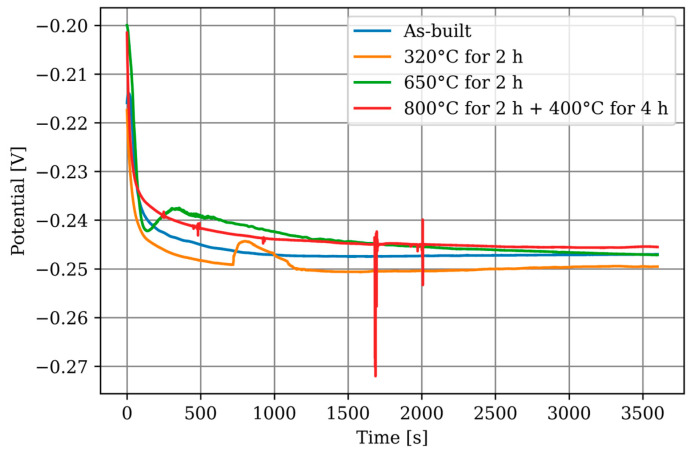
Open-circuit potential measurement over 3600 s.

**Figure 9 materials-17-03525-f009:**
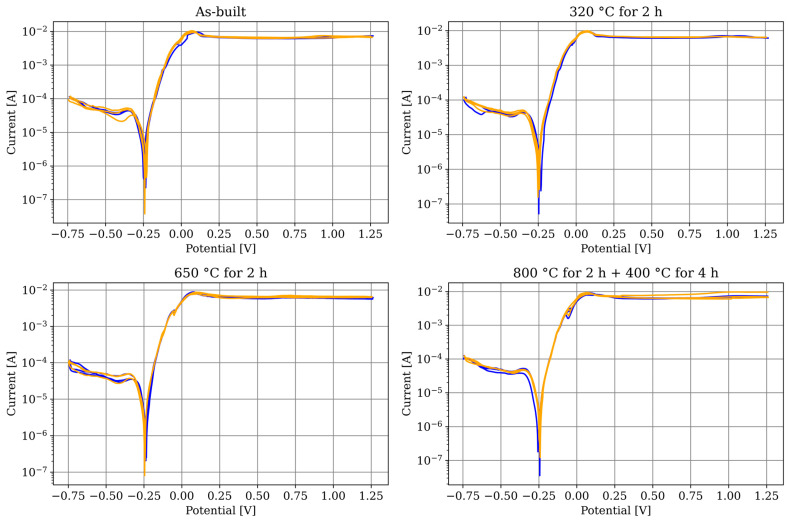
LSV measurements after 900 s of OCP measurement. The specimens in X/Y (Z) section are shown in blue (yellow).

**Figure 10 materials-17-03525-f010:**
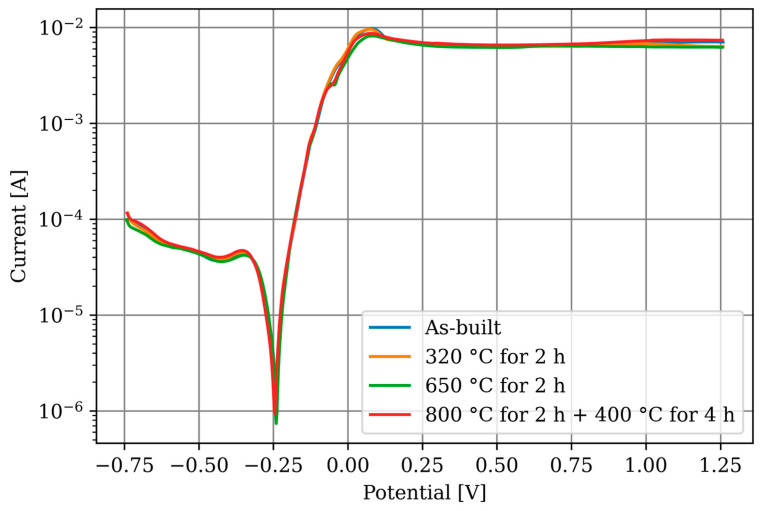
LSV measurements after 900 s of OCP measurement. The graphs are averaged from the individual measurements for each specimen.

**Figure 11 materials-17-03525-f011:**
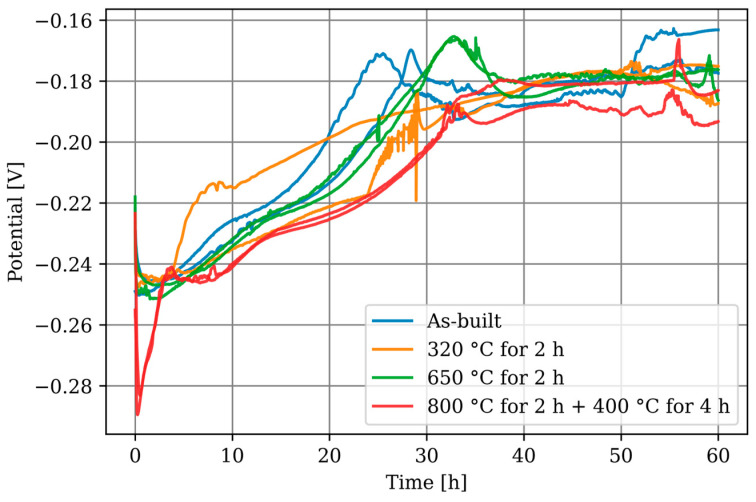
OCP measurement for 60 h.

**Figure 12 materials-17-03525-f012:**
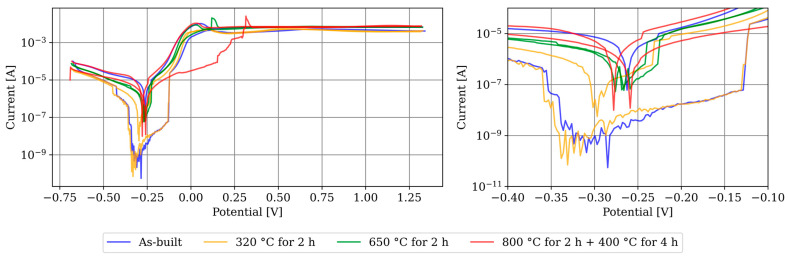
LSV measurement after 60 h of OCP. On the left, the entire graphs, and on the right, an enlargement of the transition area between the anodic and cathodic areas are shown.

**Figure 13 materials-17-03525-f013:**
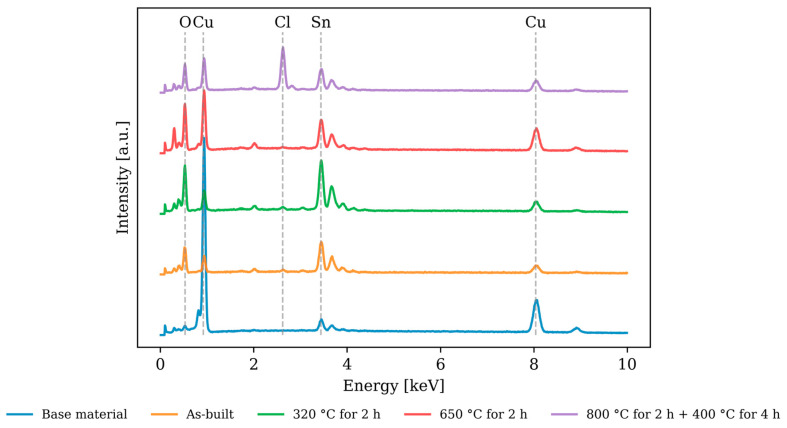
EDX spectrum of the corrosion surfaces.

**Figure 14 materials-17-03525-f014:**
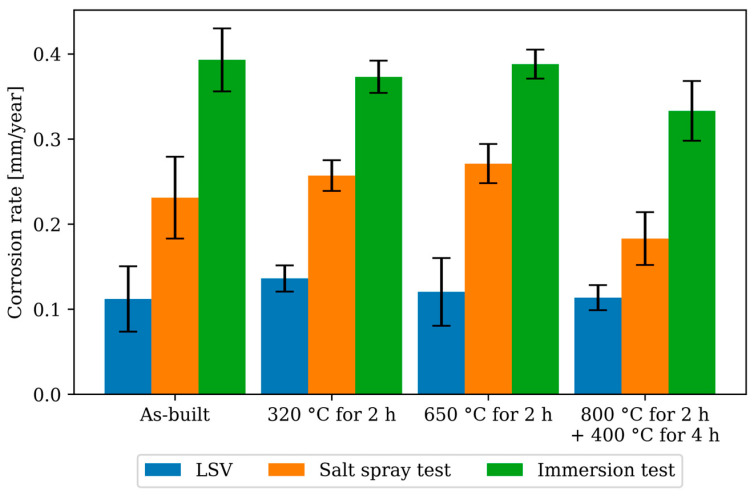
Results of the corrosion tests, LSV and immersion test in 3.5 wt.% NaCl solution at 35 °C and the salt spray test in 5 wt.% NaCl solution at 35 °C.

**Figure 15 materials-17-03525-f015:**
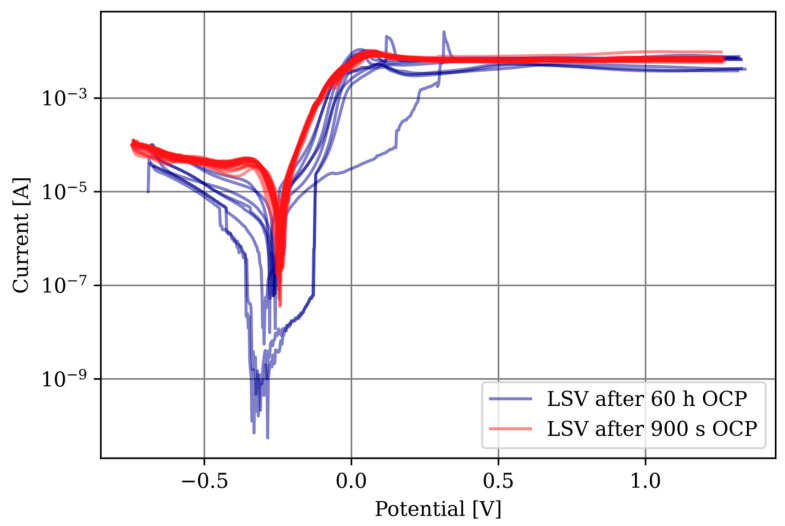
Comparison of LSV measurements after 900 s (red) and 60 h OCP measurement (blue).

**Figure 16 materials-17-03525-f016:**
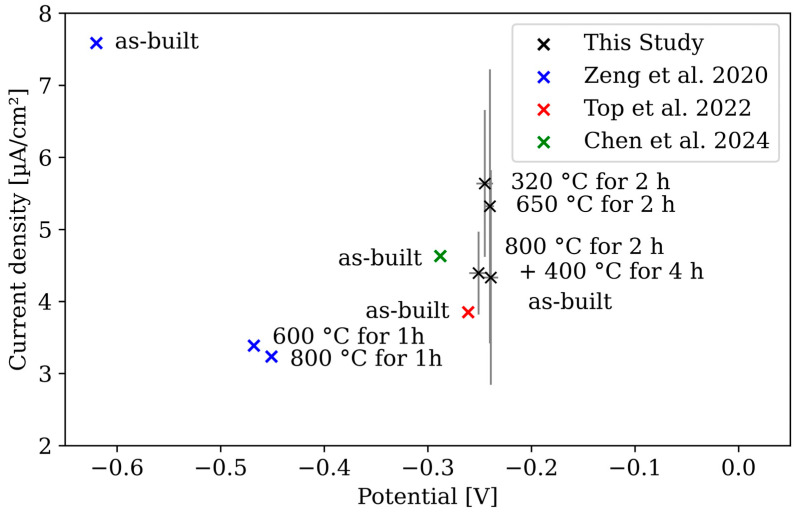
*I*_corr_ and *E*_corr_ (and standard deviation if indicated) after various heat treatments from this study and [20,24,28].

**Table 1 materials-17-03525-t001:** Parameter set used for production.

Parameter	Value	Unit
Laser power *PL*	100	W
Scan speed *v*	300	mm/s
Slice thickness *t*	0.015	mm
Hatch spacing *h*	0.06	mm
Scan strategy	Islands 5 × 5, 15° rotation	
Inert gas	Nitrogen	

**Table 2 materials-17-03525-t002:** Characteristic data from the OCP and LSV measurement as well as the calculated corrosion rate.

	Open Circuit Potential		*E*_corr_ [V]	*I*_corr_ [µA/cm^2^]	Corrosion Rate (*CR*)	
	OCP [V]	*p*-Value			*CR* [mm/Year]	*p*-Value
As-built	−0.243 ± 0.005		−0.239 ± 0.007	4.332 ± 1.490	0.112 ± 0.039	
320 °C for 2 h	−0.244 ± 0.004	0.185	−0.245 ± 0.008	5.637 ± 1.018	0.136 ± 0.015	0.959
650 °C for 2 h	−0.246 ± 0.004	0.930	−0.240 ± 0.004	5.320 ± 1.902	0.120 ± 0.040	0.262
800 °C for 2 h + 400 °C for 4 h	−0.246 ± 0.001	0.413	−0.251 ± 0.009	4.392 ± 0.575	0.113 ± 0.015	0.717

**Table 3 materials-17-03525-t003:** Chemical composition of the corrosion surfaces in atomic percent.

	Cu [at.-%]	Sn [at.-%]	O [at.-%]	Cl [at.-%]	P [at.-%]
Starting material	79.6	6.4	13.6	0.1	0.4
As-built	15.1	13.6	69.7	0.6	1.0
320 °C for 2 h	12.1	13.5	73.0	0.4	0.9
650 °C for 2 h	29.9	8.9	59.4	0.3	1.5
800 °C for 2 h + 400 °C for 4 h	19.6	9.2	58.0	12.6	0.6

**Table 4 materials-17-03525-t004:** Result of the gravimetric corrosion test.

Corrosion Rate (*CR*)	Salt Spray Test		Immersion Test	
	*CR* [mm/Year]	*p*-Value	*CR* [mm/Year]	*p*-Value
As-built	0.231 ± 0.048		0.393 ± 0.037	
320 °C for 2 h	0.257 ± 0.018	0.325	0.373 ± 0.019	0.374
650 °C for 2 h	0.271 ± 0.023	0.169	0.388 ± 0.017	0.820
800 °C for 2 h + 400 °C for 4 h	0.183 ± 0.031	0.132	0.333 ± 0.035 *	0.0450

## Data Availability

Dataset available on request from the authors.

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
