# Peer review of "Influence of Post-Heat Treatment on Corrosion Behaviour of Additively Manufactured CuSn10 by Laser Powder Bed Fusion"

_materials, 2024, doi:10.3390/ma17143525_

Round 1
Reviewer 1 Report
Comments and Suggestions for Authors
The article entitled "Influence of Post-Heat Treatment on Corrosion Behaviour and Mechanical Properties of Additively Manufactured CuSn10 by 3 Laser Powder Bed Fusion" I have found it to be highly consistent, with clearly objectives stated, with well described methods that have been used for the experiments and the assessing of the results. I have highly appreciated also the graphic illustration of the reached results and also the way that the reached results have been discussed in the "light" of results reached by other researchers which have been approached similar experiments on their research. References are all up-to-date and relevant to the approached topic of this article. The topic of realizing CuSn parts by Laser Powder Bed Fusion it is highly interesting, CuSn material being one material that is quite challenging to work with. Post-Heat Treatment methods that have been applied by the authors to improve the corrosion behaviour of the realized parts made of LBPF method are clearly described, easy to comprised and to be followed by the researchers who might aim to reproduce the way these experiments have been done in the way the authors are presenting them. Therefore based on all these arguments, my decision is to recommend the article to be accepted for publication in its current form for the MDPI Materials journal. Congratulations to the authors for their excellent work!
Author Response
Dear reviewer,
Thank you very much for your review and your kind words!
Yours sincerely
Robert Kremer
Reviewer 2 Report
Comments and Suggestions for Authors
In the presented paper, the authors analyze the influence of heat treatment on the corrosion behavior of CuSn10 tin bronze, manufactured using the additive method of Laser Powder Bed Fusion (LPBF). The LPBF method presented by the authors enables the creation of finely grained, anisotropic microstructures, whose corrosion behavior is not yet well understood – this is addressed in the reviewed paper. The authors conduct single and two-stage heat treatments of the samples, as well as hardness and microstructure analysis. Corrosion tests were performed using linear polarization, salt spray, and immersion tests.
The results obtained and presented by the authors show that heat treatments at 320 °C and 650 °C do not have a significant effect on the corrosion rate, while the two-stage treatment shows a slight improvement in corrosion resistance.
The conclusions drawn by the authors will impact the solving of various engineering problems, even in industry. The discoveries shown in the paper contribute to understanding the effects of heat treatment on the corrosion resistance of additively manufactured tin bronze and provide important insights for future applications in corrosive environments.
The paper is very well organized and contains an excellent literature review.
Please pay attention to the citation of literature on the first page – check if there is an error in line 34.
It is necessary to add a nomenclature – a full list of symbols, abbreviations, and notations to the paper. The nomenclature can be placed even at the end of the paper. However, it is worth adding it to the manuscript text.
For the material used in laboratory tests, I suggest using the word "specimen" instead of "sample," following the standards of ASTM and BS. Please check and correct the paper in this regard.
I recommend unifying the font used in the formulas throughout the paper.
It would be valuable to add a table and even a figure for the hardness measurement results, indicating the distance at which the hardness measurements were taken, how many measurements were made, the extreme values (min and max), the median, average, scatter, and standard deviation. This will demonstrate the authors' research methodology.
Units on the figures and in tables should be written in square brackets.
Overall, I have no substantive comments on the paper. It is very well written and well organized. The authors might consider improving the quality of the figures by generating them in vector format. However, this depends on the journal editor and the authors themselves.
I rate the paper very highly. I only suggest minor corrections – it is worth implementing them in the paper. I suggest a minor revision.
After making the corrections, please resend the paper for further review and acceptance.
Comments on the Quality of English LanguageMinor editing of English language required
Author Response
- Please pay attention to the citation of literature on the first page – check if there is an error in line 34.
Thank you for pointing this out. I agree with this comment. References 1 and 3 are journal articles, whereas Reference 2 is a reference book from which the specific page number was cited. For improved clarity and readability, I have removed the page number citation from Reference 2.
- It is necessary to add a nomenclature – a full list of symbols, abbreviations, and notations to the paper. The nomenclature can be placed even at the end of the paper. However, it is worth adding it to the manuscript text.
Agree. We have accordingly added a nomenclature section as Section 6 to emphasize this point.
- For the material used in laboratory tests, I suggest using the word "specimen" instead of "sample," following the standards of ASTM and BS. Please check and correct the paper in this regard.
We agree with this comment. The terminology has been revised throughout the manuscript, replacing "sample" with "specimen."
- I recommend unifying the font used in the formulas throughout the paper.
Thank you for the suggestion. All equations have been standardized to use Cambria Math, italics, size 12 font.
- It would be valuable to add a table and even a figure for the hardness measurement results, indicating the distance at which the hardness measurements were taken, how many measurements were made, the extreme values (min and max), the median, average, scatter, and standard deviation. This will demonstrate the authors' research methodology.
Thank you for this valuable suggestion. The methods section has been expanded to include details on the number of individual hardness measurements and their respective locations. Additionally, the appendix now contains a comprehensive table of the requested values.
- Units on the figures and in tables should be written in square brackets.
Agree. All units in figures and tables have been revised to be enclosed in square brackets.
Reviewer 3 Report
Comments and Suggestions for Authors
The proposed work is titled: Influence of Post-Heat Treatment on Corrosion Behaviour and Mechanical Properties of Additively Manufactured CuSn10 by Laser Powder Bed Fusion. Under my point of view the chosen title is not the best option due to that part relative to the mechanical properties in the title. The general term applied to the mechanical properties (written in this way, in plural, i.e., more than one) is much more than the study of a single mechanical property only: the hardness in the present work.
I accepted the review of the present work due to the study of the mechanical properties of a material additively manufactured by laser powder bed fusion (it was my main interest), I want to be honest, it was a disappointment to me to find that the only one mechanical property studied was the hardness.
In my opinion the interest of the work can be the same without the hardness study, because its contribution to the whole work is poor.
My suggestion is to erase the words mechanical properties of the title of the work to avoid confusion in the future potential readers.
Regarding the rest of the work, I believe that the work is well structured, well done and well written.
Kind regards.
Author Response
Dear Editor,
Thank you for giving us the opportunity to submit a revised draft of our manuscript "Influence of Post-Heat Treatment on Corrosion Behaviour of Additively Manufactured CuSn10 by Laser Powder Bed Fusion" to the Journal of Materials.
We apologise for your disappointment and fully agree with your comments and have adjusted the title according to your suggestion.
In general, we have thoroughly revised the full manuscript. All changes are highlighted in the manuscript.
Yours sincerely
Robert Kremer
Round 2
Reviewer 2 Report
Comments and Suggestions for Authors
The authors have addressed all of my previous comments in the resubmitted version of the paper.
As a result, the paper has become more valuable and readable, likely to capture the interest of readers.
They have introduced new symbols, nomenclature, clarified formulas and added figures.
Additionally, they supplemented the paper with some information.
Overall, the paper is engaging, and I have no further substantive comments to make.
Congratulations to the authors. I recommend the paper for publication.
Comments on the Quality of English LanguageMinor editing of English language required.